# Turbulence hierarchy in a random fibre laser

Iván R. Roa González[1], Bismarck C. Lima[2], Pablo I.R. Pincheira[2], Arthur A. Brum[1], Antônio M.S. Macêdo[1], Giovani L. Vasconcelos[1], Leonardo de S. Menezes[2], Ernesto P. Raposo[1], Anderson S.L. Gomes[2] & Raman Kashyap[3,4]

Turbulence is a challenging feature common to a wide range of complex phenomena. Random fibre lasers are a special class of lasers in which the feedback arises from multiple scattering in a one-dimensional disordered cavity-less medium. Here we report on statistical signatures of turbulence in the distribution of intensity fluctuations in a continuous-wave-pumped erbium-based random fibre laser, with random Bragg grating scatterers. The distribution of intensity fluctuations in an extensive data set exhibits three qualitatively distinct behaviours: a Gaussian regime below threshold, a mixture of two distributions with exponentially decaying tails near the threshold and a mixture of distributions with stretched-exponential tails above threshold. All distributions are well described by a hierarchical stochastic model that incorporates Kolmogorov's theory of turbulence, which includes energy cascade and the intermittence phenomenon. Our findings have implications for explaining the remarkably challenging turbulent behaviour in photonics, using a random fibre laser as the experimental platform.

[1] Laboratório de Física Teórica e Computacional, Departamento de Física, Universidade Federal de Pernambuco, Recife-PE 50670-901, Brazil. [2] Departamento de Física, Universidade Federal de Pernambuco, Recife-PE 50670-901, Brazil. [3] Fabulas Laboratory, Department of Engineering Physics, Polytechnique Montreal, Montreal, Quebec, Canada H3C 3A7. [4] Fabulas Laboratory, Department of Electrical Engineering, Polytechnique Montreal, Montreal, Quebec, Canada H3C 3A7. Correspondence and requests for materials should be addressed to E.P.R. (email: ernesto@df.ufpe.br).

The phenomenon of turbulence manifests itself in a myriad of natural events, such as atmospheric, oceanic and biological[1–4], as well as man-made systems including fibre lasers[5,6], Bose–Einstein condensates[7], nonlinear optics[8], and integrable systems and solitons[9,10]. In a broader context, turbulence theory has been used to explain financial market features[11,12]. In particular, atmospheric turbulence has an impact on optical communications, in which encoding of information in orbital angular momentum has been employed as a form of mitigation, in a context where Kolmogorov turbulence plays a relevant role[13].

Random lasers, predicted in the late 1960's (ref. 14), were unambiguously demonstrated in 1994 (ref. 15) and since then they have been thoroughly characterized in a diversity of systems, for example, biological materials[16], cold atoms[17], rare-earth-doped nanocrystals[18] and plasmonic-based nanorod metamaterials[19]. Among the applications, their use as speckle-free sources for imaging[20] and chemical identification[21] are some of the most promising reported so far. Random fibre lasers, on the other hand, were first demonstrated in 2007 as a quasi-one-dimensional random laser, using a photonic crystal fibre[22]. Several advances in random fibre lasers have been exploited lately, in particular in conventional optical fibres[23], including applications in optical telecommunications and temperature sensing.

In contrast to both conventional and fibre lasers, where the feedback providing gain amplification is mediated by a closed cavity formed by mirrors or fibre Bragg reflectors[24,25], the optical feedback in random lasers and random fibre lasers arises from the multiple scattering of photons in a disordered medium, thereby forming an open-complex, disordered nonlinear system, in which light propagation occurs in the presence of gain, leading to laser emission[26,27].

Recently, both random lasers and random fibre lasers have been shown to play a major role as platforms for multidisciplinary demonstrations and analogies with physical systems otherwise unavailable in the laboratory environment. Examples include astrophysical lasers[17] and statistical physics phenomena, such as Lévy statistics behaviour of intensity fluctuations[28,29], extreme events[30,31] and spin–glass analogy through the observation of the replica-symmetry-breaking phase transition[32–38]. In a recent report, Churkin et al.[39] described an approach to analyse the results of Gorbunov et al.[31] based on a modified wave kinetic model directly related to the wave turbulence scenario. More recently, a direct observation of turbulent transitions in propagating optical waves was reported[40].

In this work, we employ a theoretical framework, based on a multiscale approach to hierarchical complex systems, to explain the experimentally identified turbulent emission in the erbium-based random fibre laser (RFL) (Er-RFL) with continuous-wave (cw) pump. Turbulent behaviour in a random fibre laser is clearly demonstrated for the first time to occur both near and above the laser threshold transition. We show that the interplay of nonlinearity and disorder is essential to induce photonic turbulence in the distribution of intensity fluctuations in the random laser phase of Er-RFL. In this system, the nonlinearity arises from the process of single-photon induced nonlinear absorption, whose microscopic origin stems from the electronic levels of the erbium ions. On the other hand, the disorder is provided by the random fibre gratings inscribed in the doped fibre. In the turbulent emission regime, the theoretical description is possible only through a superposition or mixture of statistics that arises from a hierarchical stochastic model for the multiscale fluctuations.

## Results

**Theoretical framework**. A remarkable aspect of random lasers and random fibre lasers is the strong intensity fluctuations in the emission spectra, accompanied by non-trivial temporal correlations in the time series of intensity measurements, which result from the interplay between amplification, nonlinearity and disorder. Indeed, these ingredients have been shown to be essential to promote the observed shift from the Gaussian to a Lévy-like statistics of output intensities in the Er-RFL system[35].

The complex behaviour of random lasers and random fibre lasers has also been recently accounted for in a statistical physics approach that establishes a formal correspondence between these photonic systems and disordered magnetic spin glasses[32–34,41]. By starting from the Langevin dynamics equations for the amplitudes of the normal modes, a photonic Hamiltonian was obtained, which is an analogue of a class of disordered spin models. Besides the linear terms associated with the gain and radiation loss, as well as to an eventual effective damping contribution due to the cavity leakage, the Langevin equations also include a nonlinear term related to the $\chi^{(3)}$ susceptibility. The influence of the disorder mechanism manifests itself in the spatially inhomogeneous refractive index by its modulation through the $\chi^{(3)}$ susceptibility and non-uniform distribution of the gain with a random spatial profile. A rich photonic phase diagram thus emerges[32–34] as a function of the input excitation power (analogue to the inverse temperature in spin models), degree of nonlinearity and disorder strength, which tends to hamper the synchronous oscillation of the modes. In particular, in the Er-RFL system, the photonic replica-symmetry-breaking spin–glass transition was shown to coincide with the Gaussian-to-Lévy statistics shift in the distribution of output intensities[35,36]. More generally, in analogy to what happens in spin glasses, replica symmetry breaking has been shown to occur[32–38] in random lasing media, such as random lasers and random fibre lasers, which respond non-uniquely to each measurement performed under identical but time-lapsed conditions. Thus, these systems demonstrate a transition from a smooth emission below threshold, which fits a Gaussian distribution, to a Lévy statistical regime just above threshold, with strong intensity fluctuations. Hence, random lasers and random fibre lasers constitute ideal platforms to study the dynamical photonic response under changing conditions, in which a rich variety of phenomenon from chaotic behaviour to turbulence is predicted.

The introduction of nonlinearities can also give rise to wave turbulence. In ref. 42, the authors found turbulent emission in quasi-cw Raman fibre lasers in the absence of any form of built-in disorder, which was modelled by a complex Ginzburg–Landau equation. Here we employ, instead, a statistical approach to describe the non-Gaussian behaviour of the time series of intensity increments in Er-RFL, in which disorder is present in the form of customized random Bragg grating scatterers.

Our starting point is a dynamical hierarchical model recently proposed[43,44] to accommodate, through simple physical requirements, the basic concepts of Kolmogorov's statistical approach to turbulence: energy cascade, whereby energy is transferred from large to small scales, and the phenomenon of intermittency, which is the tendency of the distribution of velocity differences between two points in the fluid to develop long non-Gaussian tails. In Kolmogorov's theory, one surmises that at large Reynold's number big eddies are created spontaneously, which, because of large inertia effects, must decay into smaller eddies, in a cascade of events that go all the way down to the smallest scale, where all eddies disappear through viscous dissipation[45]. In this view, intermittency is then caused by fluctuations in the energy transfer rates or energy fluxes between adjacent scales, because otherwise (that is, if the energy fluxes

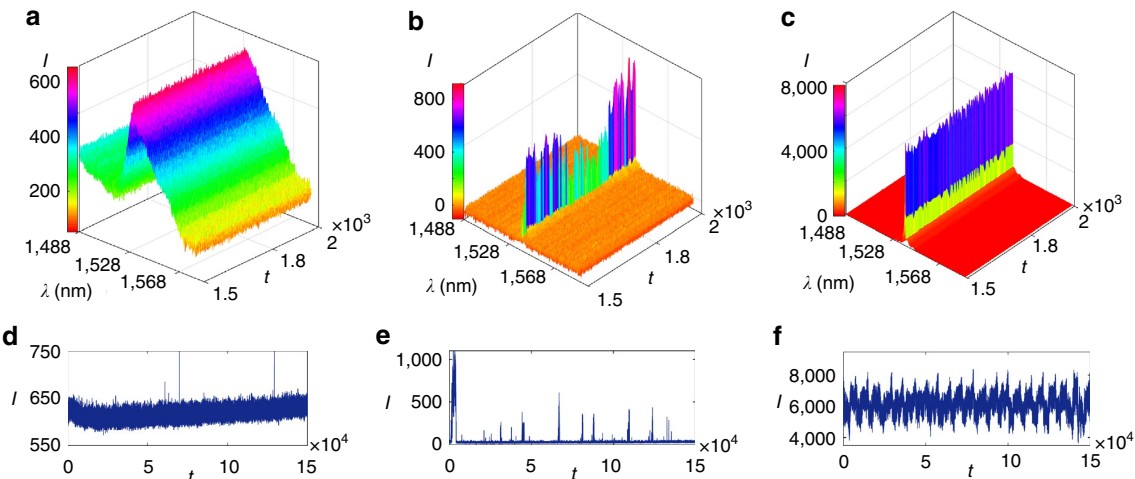

**Figure 1 | Emission spectra and time series of maximum intensities in Er-RFL.** (**a**–**c**) Representative samples showing 500 successive emission spectra at each excitation power $P$ (normalized by the threshold value $P_{th}$) below [(**a**) $P/P_{th} = 0.72$], near [(**b**) $P/P_{th} = 0.99$] and above [(**c**) $P/P_{th} = 2.92$] the random laser threshold of Er-RFL. Intensities $I$ are displayed in arbitrary units as function of the wavelength $\lambda$. The portrayed spectra were collected at times $t\tau$, with integration time window $\tau = 100$ ms and dimensionless index $t = 1,501, 1,502, \ldots, 2,000$. (**d**–**f**) For each one of the 150,000 spectra in the whole data set, the maximum intensity was recorded, resulting in the long time series $I(t)$, with $t = 1, 2, \ldots, 150,000$, which is shown in **d**–**f** for the respective values of $P/P_{th}$.

were constant) the fluid velocity fluctuations would display regular Gaussian statistics. This hierarchical model of intermittency in turbulence[43,44] will be applied below to describe non-Gaussian fluctuations in the emission intensity of Er-RFL.

By employing the cw-pumped Er-RFL system, we acquired a rather large number (150,000) of successive optical spectra for each value of the excitation power $P$, with a $\tau = 100$ ms wide integration time window. Figure 1a–c displays representative samples of 500 emission spectra for each excitation power, respectively, in the regimes below ($P/P_{th} = 0.72$), near ($P/P_{th} = 0.99$) and above threshold ($P/P_{th} = 2.92$), where the power threshold, $P_{th} = 16.30$ mW, was measured from the FWHM analysis (see Methods). We determined the maximum intensity in each spectrum of the whole data set and thus obtained a long time series of fluctuating intensity values $I(t)$, with the dimensionless time index $t = 1, 2, \ldots, 150,000$, which is illustrated in Fig. 1d–f for the respective values of $P/P_{th}$. It is clear from these plots that there is a dramatic change in the fluctuation pattern of $I(t)$ as the excitation power crosses the threshold. In fact, we shall see below that this transition actually marks the onset of turbulent emission in the Er-RFL system.

In analogy to fluid turbulence, where the relevant statistical quantities are velocity increments between two points (rather than the velocities themselves), we shall analyse here the intensity increments, $\delta I(t) \equiv I(t+1) - I(t)$, between successive optical spectra. More specifically, we define the signal associated with the intensity fluctuations as the stochastic process given by $x(t) \equiv \delta I(t)$, where var($\delta I(t)$) denotes the variance of the time series of intensity increments. If nonlinearities are not relevant, as in the prelasing regime, then the intensity increments are statistically independent and the probability distribution $P(x)$ is a Gaussian.

On the other hand, as the excitation power is increased beyond the threshold, we show below that dynamical nonlinearities give rise to turbulent emission. This implies that the Gaussian form of the signal distribution remains valid only at a local level, acquiring a slowly fluctuating variance $\varepsilon$ along the time series. In this case, we can still write its local distribution as a conditional Gaussian $P(x|\varepsilon) = \exp(-x^2/2\varepsilon)/\sqrt{2\pi\varepsilon}$, where the parameter $\varepsilon$ characterizes the local equilibrium. The basic hypothesis[46]

underlying the statistical description of turbulence is that the non-Gaussian global form of $P(x)$ can be obtained by compounding the local Gaussian with a background distribution of variance fluctuations $f(\varepsilon)$. We thus have

$$P(x) = \int_0^\infty P(x|\varepsilon)f(\varepsilon)d\varepsilon, \qquad (1)$$

where the complex dynamics (intermittency) of the turbulent state is captured by the density $f(\varepsilon)$. We take $f(\varepsilon)$ from a stochastic model that incorporates Kolmogorov's hypothesis of turbulent cascades. It has been recently shown[44] that there are two universality classes of such models, which differ with respect to the asymptotic tails of the signal distribution: one class has a power-law tail and the other shows a stretched-exponential behaviour. Here we shall describe only the stretched-exponential class, which generalizes the $K$-distribution usually employed in the description of wave scattering in a turbulent medium[47,48]. A complete presentation of the two classes can be found elsewhere[44]. We note in passing that the $K$-distribution is defined[47] by $p(x) = 2b(bx/2)^{v+1}K_v(bx)/\Gamma(1+v)$, where $K_v(x)$ is the modified Bessel function, $\Gamma(v)$ is the gamma function, and $b$ and $v$ are characteristic parameters. In particular, this distribution has exponential tails of the form $p(x) \sim x^{v+\frac{1}{2}} exp(-bx)$, for $x \gg 1$. The hierarchical distribution discussed below generalizes this behaviour to a stretched-exponential tail.

Our approach is based on a hierarchical dynamical model defined by the following set of stochastic differential equations,

$$d\varepsilon_i(t) = -\gamma_i(\varepsilon_i - \varepsilon_{i-1})dt + \kappa_i\sqrt{\varepsilon_i\varepsilon_{i-1}}dW_i(t), \qquad (2)$$

for $i = 1, \ldots, N$, where $N$ denotes the number of relevant fluctuation time scales in the background variables. Here, $\varepsilon_i$ represents the fluctuating parameters at the respective scales in the hierarchy, $\varepsilon_0$ is their long-term mean, $\gamma_i$ and $\kappa_i$ are positive constants and $W_i$ are independent Wiener processes (that is, zero-mean Gaussian processes with variance $\langle dW_i^2 \rangle = dt$). The first term in equation (2) describes the deterministic coupling between adjacent scales, which tends to cause a relaxation to the average $\varepsilon_0$, whereas the second term accounts for the multiplicative noise and is the source of intermittency. The form of the multiplicative noise is dictated by scale invariance in the

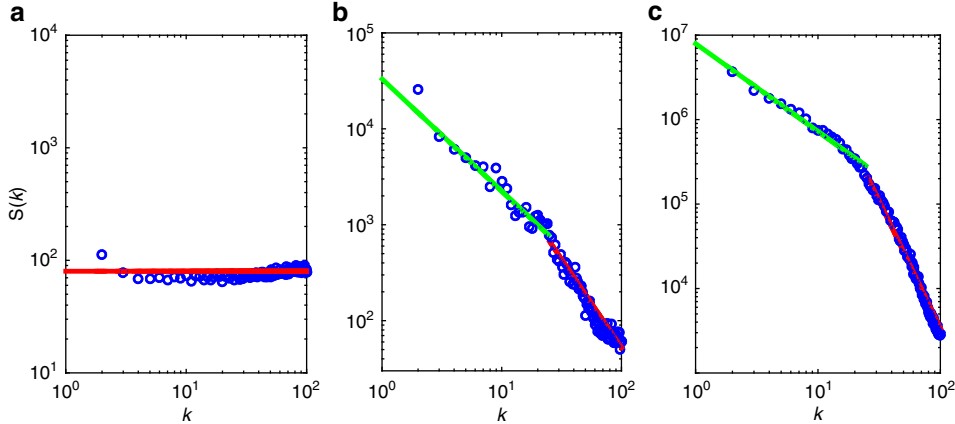

**Figure 2 | Spectral density and turbulence in the intensity dynamics in Er-RFL.** Log–log plots of the power spectral density $S(k)$ (in arbitrary units) of the time series of maximum intensities as function of the normalized frequency $k$. Results are shown for the regimes of excitation power $P$ (normalized by the threshold value $P_{th}$) (**a**) below ($P/P_{th} = 0.72$), (**b**) near ($P/P_{th} = 0.99$) and (**c**) above ($P/P_{th} = 2.92$) the threshold. Solid lines are power-law fits, $S(k) \sim k^{-\alpha}$, to the experimental data depicted in blue circles. In **a**, the white noise (horizontal red line with exponent $\alpha = 0$) is consistent with statistically independent non-turbulent Gaussian intensity fluctuations below threshold. The non-trivial double power-law behaviour, indicated by red and green lines in **b,c**, suggests the existence of turbulence in the Er-RFL dynamics both near and above the threshold. The green lines in **b,c** are associated with the exponents $\alpha = 1.17$ and $\alpha = 1.04$, respectively. The red lines in **b,c** are associated with the exponents $\alpha = 1.84$ and $\alpha = 3.01$, respectively.

sense that a rescaling of variables $\varepsilon_i \to \lambda\varepsilon_i$ should leave the dynamics unchanged[44]. Solving the stationary Fokker–Planck equation associated with equation (2), one finds that the stationary conditional probability distribution $f(\varepsilon_i|\varepsilon_{i-1})$ is given by a gamma density,

$$f(\varepsilon_i|\varepsilon_{i-1}) = \frac{(\beta_i/\varepsilon_{i-1})^{\beta_i}}{\Gamma(\beta_i)} \varepsilon_i^{\beta_{i-1}} e^{-\beta_i\varepsilon_i/\varepsilon_{i-1}}, \quad (3)$$

with $\beta_i = 2\gamma_i/\kappa_i^2$. In the regime of large separation of time scales, that is, in the case $\gamma_N \gg \gamma_{N-1} \gg \ldots \gamma_1$, we thus find that the density $f_N(\varepsilon_N)$ at the shortest scale is

$$f_N(\varepsilon_N) = \int d\varepsilon_{N-1} \ldots \int d\varepsilon_1 f(\varepsilon_N|\varepsilon_{N-1}) \ldots f(\varepsilon_1|\varepsilon_0), \quad (4)$$

where $f(\varepsilon_i|\varepsilon_{i-1})$ is given by equation (3). It can be shown that this multiple integral has a simple representation in terms of a special transcendental function, namely the Meijer $G$-function[49],

$$f_N(\varepsilon_N) = \frac{\omega}{\varepsilon_0\Gamma(\boldsymbol{\beta})} G_{0,N}^{N,0}\left(\begin{matrix} - \\ \boldsymbol{\beta} - 1 \end{matrix} \,\Big|\, \frac{\omega\varepsilon_N}{\varepsilon_0}\right), \quad (5)$$

where $\omega = \prod_{j=1}^N \beta_j$ and we have introduced the vector notation $\boldsymbol{\beta} \equiv (\beta_1, \ldots, \beta_N)$ and $\Gamma(\mathbf{a}) \equiv \prod_{j=1}^N \Gamma(a_j)$. As the first lower index of the Meijer $G$-function in equation (5) is null, the parameters in the top row are not present, as indicated by the dash[49].

The compound integral (1) for the signal distribution can thus be written as

$$P_N(x) = \frac{1}{\sqrt{2\pi}} \int_0^\infty \exp\left(-\frac{x^2}{2\varepsilon_N}\right) \varepsilon_N^{-1/2} f_N(\varepsilon_N) d\varepsilon_N, \quad (6)$$

where $f_N(\varepsilon_N)$ is given by equation (5). Using a convolution property of the $G$-function[49], this integral can be performed explicitly, yielding

$$P_N(x) = \frac{\omega^{1/2}}{\sqrt{2\pi}\varepsilon_0\Gamma(\boldsymbol{\beta})} G_{0,N+1}^{N+1,0}\left(\begin{matrix} - \\ \boldsymbol{\beta} - 1/2, 0 \end{matrix} \,\Big|\, \frac{\omega x^2}{2\varepsilon_0}\right). \quad (7)$$

As anticipated above, for $N = 1$ this expression recovers the $K$-distribution. The large-$x$ asymptotic limit of equation (7) is a

modified stretched exponential[44],

$$P_N(x) \sim x^{2\theta}\exp\left[-(N+1)(\omega x^2/2\varepsilon_0)^{1/(N+1)}\right], \quad (8)$$

where $\theta = (\sum_{i=1}^N \beta_i - N)/(N+1)$.

Finally, a more general family of distributions can be obtained when the data contain some internal structure that could lead to the presence of clusters of statistically independent samples. In this case, we may decompose $P_N(x)$ as a discrete statistical mixture of multiscale distributions,

$$P_N(x) = \sum_{j=1}^n p_j P_N^{(j)}(x), \quad (9)$$

where the statistical weights $p_j$ satisfy $\sum_{j=1}^n p_j = 1$ and $P_N^{(j)}(x)$ is obtained from equation (7). This form of discrete statistical mixture has found applications, for example, in oceanic turbulence[50] and finance[51].

**Experimental results.** The Er-RFL fabrication along with the fibre Bragg grating inscription is detailed in ref. 52 and the experimental setup is the same as reported in refs 35,36 (see Methods).

As a first quantitative characterization of the experimental time series $I(t)$ of maximum intensities, shown in Fig. 1d–f, we calculated the power spectral density,

$$S(k) = \left|\sum_{n=1}^L I_n \exp\left(\frac{-2\pi i(n-1)(k-1)}{L}\right)\right|^2, \quad (10)$$

by subdividing the time series into 550 windows of size $L = 256$, evaluating $S(k)$ for each window and then performing the average of $S(k)$ over all windows. Here, $I_n$ denotes the $n$th value of the intensity inside the window and $k/L$ is the dimensionless frequency. In Fig. 2 we show log–log plots of $S(k)$ for the three mentioned values of the excitation power: (a) $P/P_{th} = 0.72$, (b) $P/P_{th} = 0.99$ and (c) $P/P_{th} = 2.92$, corresponding, respectively, to the regimes below, near and above the threshold. The lines are fits to the power-law behaviour $S(k) \sim k^{-\alpha}$. The white noise ($\alpha = 0$) observed below threshold in Fig. 2a is consistent with the statistically independent non-turbulent intensity fluctuations,

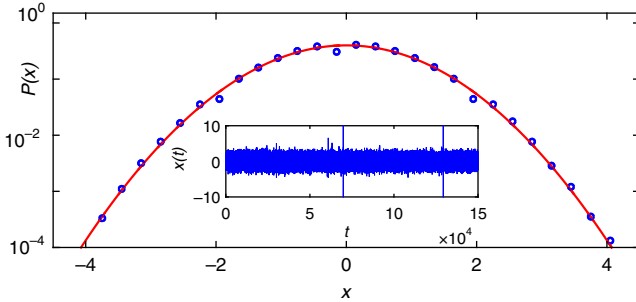

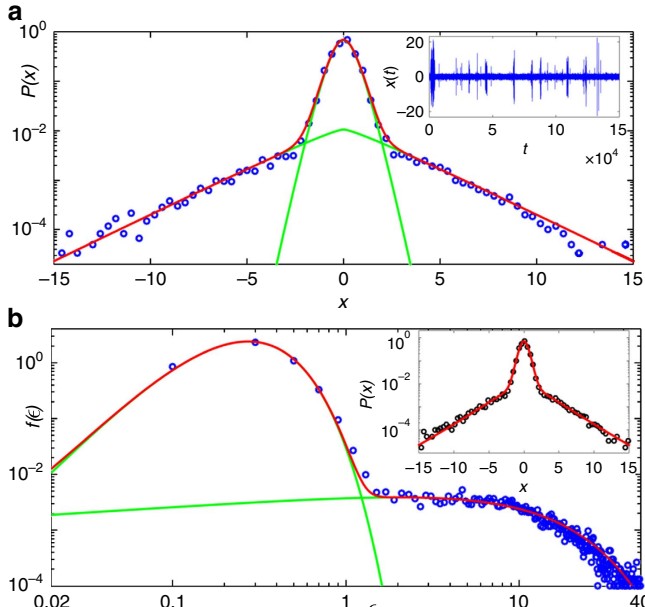

**Figure 3 | Gaussian distribution of intensity increments and non-turbulent prelasing behaviour in Er-RFL.** Semi-log plot of the distribution $P(x)$ of experimental intensity increments $x$ in the non-turbulent prelasing regime at the excitation power (normalized by the threshold value) $P/P_{th} = 0.72$ (blue circles) and best fit to a Gaussian function (red line). The intensity increments are defined as the stochastic process given by $x(t) \equiv \delta I(t)$, where $\delta I(t) \equiv I(t+1) - I(t)$ denotes the fluctuations of maximum intensities in 150,000 successive optical spectra and var($\delta I(t)$) is the variance of the time series of intensity increments. The inset displays the corresponding time series of $x(t)$ as function of the dimensionless time index $t = 1, 2, \ldots, 150,000$.

**Figure 4 | Statistical mixture of intensity increments and turbulent behaviour near the threshold in Er-RFL.** (**a**) Semi-log plot of the distribution $P(x)$ of experimental intensity increments $x$ (blue circles) near the threshold, at the excitation power (normalized by the threshold value) $P/P_{th} = 0.99$ and the prediction from the hierarchical model (2) for the statistical mixture (11) (red line). The individual components of the mixture, given by two $K$-distributions with $N = 1$, are shown by green lines. The inset shows the corresponding time series of intensity increments. A single time scale ($N = 1$) in the dynamics of the background variable characterizes the turbulent behaviour of the intensity fluctuations dynamics in Er-RFL near the threshold. (**b**) Log–log plot of the density function $f(\epsilon)$ of the experimental variance series $\epsilon(t)$ (blue circles) and the hierarchical model prediction for the statistical mixture (12) (red line), with same parameters as in **a**. The individual components of the statistical mixture are shown by green lines. The inset displays the compounding of $\epsilon(t)$ with a Gaussian function (red line) and the experimental distribution (black circles).

displaying Gaussian distributions for both intensities and intensity increments.

In contrast, the non-trivial double power-law behaviour, noticed both near and above threshold in Fig. 2b,c, respectively, suggests the existence of turbulence in the intensity fluctuations dynamics, which is confirmed below. Such behaviour of $S(k)$ is also consistent with the double cascade phenomenon observed in the wave turbulence scenario[53].

In Fig. 3 we display the experimental distribution of intensity increments below threshold, at $P/P_{th} = 0.72$. The good Gaussian fit corroborates the results for the prelasing regime in Fig. 2a. The variance $\varepsilon$ does not fluctuate, corresponding to the density $f(\varepsilon)$ given by a Dirac delta function in the compound integral (equation (1)). Consequently, a multiscale turbulent cascade is absent below threshold ($N = 0$ in the theoretical model), which implies that the dynamics of intensity fluctuations in Er-RFL produces intensity increments that are statistically independent.

The scenario is drastically modified as the excitation power is increased near and above the threshold. Indeed, the excellent fit to the distribution of intensity increments observed in Fig. 4(a) at $P/P_{th} = 0.99$ is described by the statistical mixture

$$P_N(x) = p P_N(\beta, \varepsilon_0; x) + (1 - p) P_N(\beta', \varepsilon_0'; x), \quad (11)$$

where $P_N(\beta, \varepsilon_0; x)$ is given by equation (7), with $\beta_j = \beta$ for all $j$. The values of the parameters are $N = 1$ ($K$-distribution), $p = 0.94$, $\beta = 4.21$, $\varepsilon_0 = 0.36$, $\beta' = 1.2$ and $\varepsilon_0' = 12.0$. In this case, the statistical model predicts a single time scale ($N = 1$) influencing the dynamics of the background variable (that is, the fluctuating variance of intensity increments), which in turn characterizes the turbulent behaviour of the fluctuation dynamics of the output intensity in Er-RFL near the threshold.

To verify the compound hypothesis expressed in equations (1) and (6), we implemented a procedure to compute a subsidiary time series of variance estimators $\epsilon(t)$. To this end, the time series $x(t)$ of intensity increments was subdivided into overlapping intervals of size $M$ and for each such interval we computed the variance estimator $\epsilon(t) = \frac{1}{M} \sum_{j=1}^{M} [x(t - j) - \bar{x}(t)]^2$, where $\bar{x}(t) = \frac{1}{M} \sum_{j=1}^{M} x(t - j)$ and $t = M, \ldots, 150000$, thus generating a new time series. Next, we numerically compounded the distribution of $\epsilon(t)$ with a Gaussian function, as suggested by equation (6), for various $M$ and selected the value of $M$ for which

the corresponding superposition integral best fitted the distribution of the original time series. Excellent agreement for $P(x)$ was found using $M = 15$, as seen in the inset of Fig. 4b. The optimal value of $M$ can be interpreted as an estimation of the large timescale associated with fluctuations in the background variable $\varepsilon$. In the main plot of Fig. 4b, we show the good agreement between the distribution related to the $\epsilon(t)$-series generated using the optimal window size $M$ and the background density $f_N(\varepsilon_N)$ with statistical mixture

$$f_N(\varepsilon_N) = p f_N(\beta, \varepsilon_0; \varepsilon_N) + (1 - p) f_N(\beta', \varepsilon_0'; \varepsilon_N), \quad (12)$$

where $f_N(\beta, \varepsilon_0; \varepsilon_N)$ is given by equation (5), for the same parameters as in Fig. 4a, as expected from the consistency with the joint fit of statistical mixtures. The individual components of the mixture are shown by green lines in both Fig. 4a,b.

Figure 5a shows the distribution of intensity increments well above the threshold, at $P/P_{th} = 2.92$. The red line displays the fit to a statistical mixture in the form of equation (11), with parameters $N = 6$, $p = 0.30$, $\beta = 8.3$, $\varepsilon_0 = 0.19$ and $\varepsilon_0' = 1.3$. The background distribution is shown in Fig. 5b, along with the fit (red line) to the statistical mixture as in equation (12), with the same parameters. Excellent agreement is found in both fits. The superposition of the distribution of the variance estimator ($M = 22$) with a Gaussian distribution is depicted by the red line in the inset of Fig. 5b. The individual components of

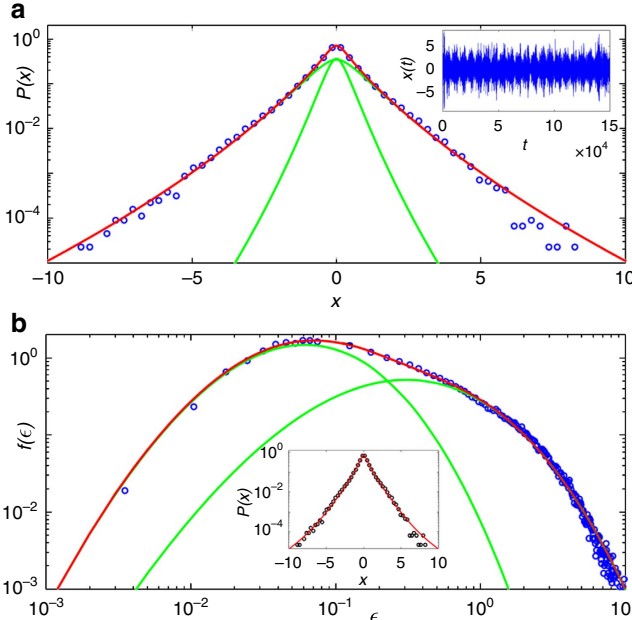

**Figure 5 | Statistical mixture and multiscale turbulent behaviour above the threshold in Er-RFL.** (**a**) Semi-log plot of the distribution $P(x)$ of experimental intensity increments $x$ (blue circles) in the regime well above threshold, at the excitation power (normalized by the threshold value) $P/P_{th} = 2.92$ and the prediction from the hierarchical model (2) for the statistical mixture (11) (red line). The individual components of the mixture, given by two Meijer $G$-distributions with $N = 6$, are shown by green lines. The expected rise in the number of relevant time scales ($N = 6$) upon increasing the excitation power is consistent with the multiscale turbulent cascade behaviour of the intensity fluctuations dynamics in Er-RFL above threshold. The inset shows the corresponding time series of intensity increments. (**b**) Log–log plot of the density function $f(\epsilon)$ of the experimental variance series $\epsilon(t)$ (blue circles) and the hierarchical model prediction for the statistical mixture (12) (red line), with same parameters as in **a**. The individual components of the statistical mixture are shown by green lines. The inset displays the compounding of $\epsilon(t)$ with a Gaussian function (red line) and the experimental distribution (black circles).

the mixture are shown by green lines in both Fig. 5a,b. We observe that the changes in the parameters $\beta$, $\beta'$, $\epsilon_0$, $\epsilon_0'$ and $p$ at $P/P_{th} = 2.92$, in comparison with the respective values near the threshold, accommodate well the changes in the shapes of the experimental distributions for both $x(t)$ and $\epsilon(t)$ caused by an enhancement of the intermittency effect. For example, the decrease in the weighting parameter $p$ (from $p = 0.94$ at $P/P_{th} = 0.99$ to $p = 0.30$ at $P/P_{th} = 2.92$) indicates an increase in the statistical relevance of the underlying structure associated with the higher mean variance $\epsilon_0'$, whose weight is $(1 - p)$, which is reasonable, as fluctuations become stronger above the threshold. Similarly, the observed value of $N$ is also consistent with the expected rise in the number of relevant time scales upon increasing the excitation power.

In this respect, we also notice that the emergence of $K$-distributions in weak-scattering by continuous media, such as sea clutter, has been attributed[54] to the modulation of small-scale fluctuations by more slowly changing large-scale structures, a scenario that is consistent with our hierarchical model with a single timescale for the background ($N = 1$). In Er-RFL near the threshold, a similar mechanism may develop, whereby large structures consisting of groups of correlated scatterers may (intermittently) form, thus yielding $N = 1$, as verified above. Beyond the threshold, stronger nonlinearities may lead to

multiscale dynamics where small-scale intensity fluctuations are modulated by larger scales, which, in turn, are modulated by even larger scales and so on, up to the largest scale in the system. This cascade-like behaviour thus implies the existence of a number $N > 1$ of characteristic scales—a view that is precisely captured by the hierarchical model (2)— with $N$ expected to grow as the excitation power increases, provided there is no saturation of the gain or the nonlinearity. We remark, however, that although the existence of a turbulence hierarchy in the Er-RFL system has been clearly identified from a statistical analysis of a large set of emission spectra, more studies are necessary to develop a comprehensive understanding of the underlying physical mechanisms.

## Discussion

From the analysis above, we observe that in the regimes near and above the threshold, where the roles of disorder and nonlinearity are mostly evidenced in the Er-RFL system, the theoretical description is possible only through a mixture of statistics that arises from a hierarchical stochastic model for the multiscale intensity fluctuations, which is strictly related to the emergence of Kolmogorov's turbulence behaviour in the distribution of intensity increments.

In this sense, the extensive size of the experimental data set was proved essential to unveil the turbulent emission behaviour in Er-RFL. Indeed, from the discussion above we infer that a significant analysis of the variance fluctuations in the compound integral should require at least $\sim 10^5$ emission spectra.

We should however point out that discrete statistical mixtures are usually associated with a partition of the data into statistically independent subsets. In ref. 50, for instance, the probability density function of the dissipation rate of kinetic energy in oceanic turbulence has been found to be bimodal and well described by a mixture of two log-normal distributions. In this case, the authors speculate that the partition of the data was due to a combination of an active mode and a quiescent one. Although we did not attempt to separate our experimental data according to some identified mechanism responsible for the statistical mixtures observed both near and above the threshold, we can infer from general grounds that such mechanism could arise from a subtle combination of stimulated and spontaneous turbulent emissions in the presence of both nonlinearity and disorder. A detailed quantitative description of this particular issue is, however, beyond the scope of the present study and will be subject of a future investigation.

In conclusion, we reported on the first observation of the statistical signatures of turbulent emission in a cw-pumped one-dimensional random fibre laser, with customized random Bragg grating scatterers. The distribution of intensity increments in an extensive data set exhibits three qualitatively different forms as the excitation power is increased: it is Gaussian below threshold, it behaves as a statistical mixture of $K$-distributions near the threshold and it is well described by a mixture of Meijer $G$-distributions with a stretched-exponential tail above threshold. A recently introduced hierarchical stochastic model[43,44], consistent with Kolmogorov's theory of turbulence, was used to interpret the experimental data. It is also a striking fact that the emergence of turbulence behaviour coincided precisely with the onset of the photonic replica-symmetry-breaking spin–glass phase at the laser threshold in Er-RFL[35,36]. In fact, we observe that amplification, nonlinearity and disorder are the essential ingredients to induce both the photonic glassiness and turbulent phenomena in this system. It remains, however, rather elusive whether or not there exists some strict interplay between these properties mediated by the common underlying mechanisms.

For instance, although these ingredients also constitute the basis to the glassy properties and Lévy statistics of intensity fluctuations that have been concurrently demonstrated[35–38] in some random lasers and random fibre lasers, it has been recently shown[55] that a rigorous connection between these features is not mandatory, so that there can be circumstances in which, for example, a glassy phase emerges along with a Gaussian statistics of intensity fluctuations. We thus hope that our work stimulates this unique opportunity to further investigate on these remarkably challenging complex phenomena through controlled photonic experiments in random lasers and random fibre lasers.

## Methods

**Er-based random fibre laser.** The Er-RFL fabrication, including the fibre Bragg grating inscription, is detailed in ref. 52. It employs a polarization maintaining erbium-doped fibre from CorActive (peak absorption $28 \, dB \, m^{-1}$ at 1,530 nm, numerical aperture (NA) = 0.25, mode field diameter 5.7 μm), in which a randomly distributed phase error grating was written. Using this procedure, a very high number of scatterers ($\gg 10^3$) was implemented, improving the fibre randomness. A fibre length of 30 cm was used in the present work. The measured threshold from the analysis of the full width at half maximum (FWHM) was $P_{th} = (16.30 \pm 0.05)$ mW. The Er-RFL linewidth was limited by our instrumental resolution to 0.1 nm. We remark that the number of longitudinal modes in the Er-RFL, measured using a speckle contrast technique, is ~204 (ref. 36). This finding corroborates the multimode character of the Er-RFL system.

**Intensity measurements.** For the intensity fluctuations measurements, an extensive sequence of 150,000 emission spectra was collected for each excitation power in the regimes below, near and above threshold. A home-assembled semiconductor laser operating in the cw regime at 1,480 nm was used as the pump source. The Er-RFL output was directed to a 0.1 nm resolution spectrometer with a liquid-N$_2$ charge-coupled device camera sensitive at 1,540 nm. The spectra for each power were acquired with integration time $\tau = 100$ ms. We stress that the intensity fluctuations of the pump source, <5%, were not correlated with the fluctuations analysed here, as pointed out in refs 31,34, and also specifically in the present experimental setup through the measurement of the normalized s.d. of both the pump laser and the Er-RFL system. Indeed, although this quantity remained constant in the former, it varied substantially in Er-RFL (see ref. 35).

**Data availability.** All relevant data are available from the authors.

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

## Acknowledgements

We acknowledge Conselho Nacional de Desenvolvimento Científico e Tecnológico—CNPq, Fundação de Amparo à Ciência e Tecnologia do Estado de Pernambuco—FACEPE (Núcleo de Excelência em Nanofotônica e Biofotônica—Nanobio and Núcleo de Excelência em Modelagem de Processos e Fenômenos Físicos em Materiais e Sistemas Complexos), and Instituto Nacional de Fotônica—INFo (Brazilian agencies).

## Author contributions

I.R.R.G., A.M.S.M. and G.L.V. performed the theoretical study. I.R.R.G. and A.A.B. did the numerical analysis. R.K. designed and fabricated the erbium-based random fibre laser with random Bragg grating scatterers. B.C.L., P.I.R.P., L.d.S.M., E.P.R. and A.S.L.G. conceived the experimental study. B.C.L. and P.I.R.P. performed the experiments. I.R.R.G., A.M.S.M., G.L.V., L.d.S.M., E.P.R. and A.S.L.G. discussed the results and wrote the manuscript.

## Additional information

**Competing interests:** The authors declare no competing financial interests.

