## [Peer Review File · Nature Communications]

Reviewers' comments:

Reviewer #1 (Remarks to the Author):

Review of the manuscript entitled "Turbulence Hierarchy in a Random Fiber Laser" by González and coworkers.

In this work, the authors use a dynamical hierarchical model recently proposed by some of them (arXiv of Ref. [43]) to explain the turbulent emission observed in cw-pumped erbium-based random fiber lasers. The experimental data acquired below, near and above lasing threshold are all very well described by this formalism, thereby constituting a solid theoretical framework for the study of turbulent behaviors in such systems.

I am not aware of similar studies in the literature. The theoretical formalism is physically sound and - as far as I have seen - free from errors. The experimental methods and data analysis appear to be valid. The conclusions of the work are well supported.

The theoretical formalism is presented in general terms in Ref. [43] (arXiv paper deposited in January 2017), but it has not been applied to analyze random lasing processes. The fact that it works well brings new insight onto the physical mechanisms underlying lasing in random fiber lasers. I am rather convinced that the paper will be of great interest to the community and that it might impact future research.

For these reasons, I recommend the publication of this manuscript in Nature Communications.

I would have found it useful and interesting to provide a phenomenological (qualitative) explanation of the changes observed in the fitted parameter values corresponding to the distribution functions of Figs. 3(a) and 4(a). It is written that the increase of N "is consistent with the expected raise in the number of relevant time scales upon increasing the excitation power" but what about the mixing ratios, the beta factors that increase, the variance fluctuation of the largest time scale?

Reviewer #2 (Remarks to the Author):

The manuscript targets an area of great interest: It has become quite clear in recent years that lasers and in particular fiber lasers provide one of the best experimental platforms for studies of non-equilibrium statistical physics (as well as fluid dynamics) as a result of relative ease of quantitative analysis, easy control of various factors, such as strength of fluctuations and nonlinear interactions. So, from this general point of view, the present manuscript is highly welcome.

However, this manuscript fails to present its results in a manner that is accessible by a broad audience. In its current form, it would be appropriate, style-wise, for Phys. Rev. A or E, perhaps, but certainly it is not understandable by a general audience. These comments are purely about the presentation of the manuscript and not at all about its contents. I consider myself experienced in various aspects covered by this manuscript and I had to reread many sentences multiple times to understand what was being said. Unfortunately, this extends even to the figures. At the level of Nature Comm. or similar broad-interest journals, I believe it is expected that one should catch good portion of the manuscript's message by reading the abstract and the figures, together with their captions. This is one reason why many papers have rather long captions. In this case, even though I read the manuscript in parallel, I had to work hard to figure out what was being plotted. I believe it really has to be rewritten with very substantial changes in the narration for it to become appropriate for Nature Communications. Much of what I understood, I understood after considerable mental effort and after consulting several of their references. This should not be the case, especially for a journal with broad readership.

Regarding the scientific content, my view is that the results are interesting and certainly represent a significant progress and should be published in Nature Communications, particularly if the physical interpretation and clarity can be improved. OK, we have learned that there are three main, qualitatively different regimes of intensity fluctuations, below, around and far above the threshold. We also have learned that the observed regimes, in particular the last one, can be explained with statistical distributions that can be interpreted as arising from a hierarchy of multiple time scales. This last bit is very interesting. But, then, the story ends. At least this episode ends. The authors tell us wait for the next episode, where they will attempt a more detailed analysis to be presented in another paper. Thus, in addition to revising the text to improve clarity, the authors are invited to extend on the physical interpretation and implications of the observations being reported here. I do not think the authors need to do additional work, but explain (much) better. I expect that after such a revision, the manuscript should be suitable for publication in Nature Communications.

Below, I will outline some specific points that relate to the clarity of the message or some minor technical points. They are presented in no particular order:

- Part of the difficulty arises from the language. Please just simplify it. Most of the readers are not native speakers.
- The manuscript should be more self-contained. E.g., what is a K-distribution? It is unreasonable to assume every reader knows it. Wiener is better known, but even that still requires some holding of the reader's hand.
- Some of the terminology is strange for an optics/lasers person: "Intensity spectra" appears without being well defined. If they had said optical spectra, it would have been more understandable. I initially thought that they recorded the intensity over time and then calculated its spectrum (as you would do to analyze, e.g., intensity/amplitude noise). I had to look up the JOSA B paper of the authors to be sure about it. I would simply include an figure showing an example optical spectrum, just like their Figure 2 in the JOSA B paper (reference 35 of the present manuscript).
- The plots need also be more clear. Fig. 2, main figure, vertical axis. It is logarithmic, but there is only one tick mark. There is none other (same problem in Figure 3a). So the scale is unspecified. The same figure's inset: it is the same thing in linear scale, but the data is plotted as bars. This does not match the main figure and confused me before I figured it out from the caption.
- Same figure, caption: What is an intensity increment? Yes, OK, I understood it eventually, but please explain things clearly or use more common terminology.
- The potentially interesting remark about the spin-glass paper (reference 36) is not clear. I still do not understand it.
- A technical comment: The pump noise is specified to be less than 5% (how much less? we do not know, but presumably not much less than 5% otherwise the authors would have said so) means that at least for 30-40% of the time the laser is above threshold (for the case of $P/P_{th} = 0.99$), since P itself varies by ± 0.05 . So, the laser randomly goes above and then falls below the laser threshold. Surely, this must have consequences ($N=2?$). The authors should remark.
- Can the authors comment on the physical interpretation of the hierarchy number, N ? $N = 2$, perhaps (probably) naively, can be imaged as corresponding to the two time scales associated with the laser being (occasionally) above and then below threshold (related to the previous comment). As P/P_{th} is further increased, N also increases to 6, indicating increasingly complex behavior, developing turbulence. Well, how does N vary as P is increased? Does it increase rapidly to 6 and

more or less stay there? What would N be if P/P_{th} was even larger? Does it saturate? Any comments would be appreciated.

RESPONSE TO REFEREES

Re: Manuscript "Turbulence hierarchy in a random fiber laser", by I. R. R. González et al. Code NCOMMS-17-02149.

* RESPONSE TO THE REVIEWER #1

We thank Reviewer #1 for the careful and thorough reading of our manuscript. We are pleased to read in his/her report that "I am rather convinced that the paper will be of great interest to the community and that it might impact future research."

Reviewer #1 made some interesting remarks which we carefully addressed in the revised manuscript, as also detailed below. As a consequence, we think the manuscript is improved significantly both in the presentation, in order to make it more accessible to a wide readership, and in the physical discussion of the results.

We therefore hope that, with these improvements, the revised version of the manuscript can now be considered suitable for publication.

1. COMMENT:

> I would have found it useful and interesting to provide a phenomenological (qualitative) explanation of the changes observed in the fitted parameter values corresponding to the distribution functions of Figs. 3(a) and 4(a). It is written that the increase of N "is consistent with the expected raise in the number of relevant time scales upon increasing the excitation power" but what about the mixing ratios, the beta factors that increase, the variance fluctuation of the largest time scale?

1. ANSWER: We agree with this nice suggestion and added the required explanation in the paragraph after Eq. (12) of page 5 of the revised manuscript. For example, we remark that the shift in the weighting parameter in Eq. (11), from $p = 0.94$ near the threshold to $p = 0.30$ at $P/P_{th} = 2.92$, indicates an increase in the statistical relevance of the component of the mixture that is associated with the largest mean variance, a result that is expected since fluctuations become stronger above the threshold.

We also comment that the changes in the remaining parameters (increase of β and β' , decrease of ϵ_0 and ϵ_0'), as the excitation power is raised, accommodate well the changes in the shapes of the experimental distributions for both $x(t)$ and $\epsilon(t)$ caused by an enhancement of the intermittency effect.

Moreover, we also added a whole new paragraph (penultimate paragraph of page 5 starting with "In this respect...") that deepens the discussion on the rising number of relevant time scales upon increasing the excitation power. Indeed, near the threshold the emergence of the K-distribution ($N = 1$) may be attributed to the modulation of small-scale fluctuations by more slowly changing large-scale structures. Beyond the threshold, however, stronger nonlinearities may lead to a multiscale dynamics where small-scale intensity fluctuations are modulated by larger scales, and so on up to the largest scale, in

a turbulent-cascade-like scenario. The value of N thus represents an estimate of the number of relevant levels in this cascade. In this context, a new reference has been added (ref. [54]) where a similar discussion is provided about the emergence of K -distributions in weak-scattering by continuous media.

2. COMMENT:

> The theoretical formalism is presented in general terms in Ref. [43] (arXiv paper deposited in January 2017), but it has not been applied to analyze random lasing processes. The fact that it works well brings new insight onto the physical mechanisms underlying lasing in random fiber lasers.

2. ANSWER: We have updated this reference (current ref. [44] in the revised manuscript), which now appears published in the Physical Review E. We have also added ref. [43] in which the general model was first proposed, including its connection with the turbulence phenomenon.

RESPONSE TO REFEREES

Re: Manuscript “Turbulence hierarchy in a random fiber laser”, by I. R. R. González et al. Code NCOMMS-17-02149.

* RESPONSE TO THE REVIEWER #2

We thank Reviewer #2 for the careful reading of our manuscript and for the several valuable suggestions provided. His/Her statement that “(...) from this general point of view, the present manuscript is highly welcome” sounds quite motivating for us.

Reviewer #2 was mostly concerned about the need for improvement in the presentation of the manuscript (both text and figures), in order to make it more accessible to a broad readership, and also about the deepening of the physical discussion of our results at certain specific points. We entirely agree with Reviewer #2 and have made a considerable effort to improve the manuscript in these two aspects.

As detailed below, each one of the several relevant points raised by Reviewer #2 was carefully addressed in the revised manuscript. We therefore hope that, with these improvements, the manuscript can now be considered suitable for publication.

1. COMMENT:

> However, this manuscript fails to present its results in a manner that is accessible by a broad audience. In its current form, it would be appropriate, style-wise, for Phys. Rev. A or E, perhaps, but certainly it is not understandable by a general audience. These comments are purely about the presentation of the manuscript and not at all about its contents. I consider myself experienced in various aspects covered by this manuscript and I had to reread many sentences multiple times to understand what was being said.

1. ANSWER: We regret that the desirable excellence in clarity for a general audience could not be achieved already in the original version of the manuscript. We thus worked hard to overcome this problem in the revised manuscript.

To this end, we rephrased several sentences to improve clarity and also introduced additional pieces of information intended to make the underlying concepts and definitions clearer to non-experts. Some examples include the justification for the use of intensity increments (instead of the maximum intensities themselves) in the analysis of photonic turbulence (penultimate paragraph of page 2), the introduction of some basic remarks on the K-distribution (text after Eq. (1) of page 3), the definition of intermittency in the context of turbulence (fourth paragraph of the Results section on page 2), a more comprehensive description of the relevant quantities in Eq. (2), new texts improving the connection with the replica-symmetry-breaking spin-glass phase observed above threshold (second paragraph of the Results section and concluding paragraph), as well as a great number of minor changes throughout the manuscript (see also below).

2. COMMENT:

> Unfortunately, this extends even to the figures. At the level of Nature Comm. or similar broad-interest journals, I believe it is expected that one should catch good portion of the manuscript's message by reading the abstract and the figures, together with their captions. This is one reason why many papers have rather long captions. In this case, even though I read the manuscript in parallel, I had to work hard to figure out what was being plotted. I believe it really has to be rewritten with very substantial changes in the narration for it to become appropriate for Nature Communications. Much of what I understood, I understood after considerable mental effort and after consulting several of their references. This should not be the case, especially for a journal with broad readership.

2. ANSWER: Besides re-examining the text very carefully, we also took special care of the figures in the revised manuscript. In each one of them, we analyzed minutely if the complete information is readily available to the reader (including tick marks, axes, symbols, units, labels, etc.), and also discussed the best way to present them (e.g., Fig. 3 of the revised manuscript now appears more easily readable after the removal of one of the insets that actually did not add any new information; see also our response in item 7 below). Much attention was also given to improve the figure captions, in the sense commented above by Reviewer #2.

Moreover, following the suggestion by Reviewer #2, we also added a new figure (Fig. 1 in the revised manuscript; see also our response in item 6 below) that will certainly help to make our results clearer and more easily understandable to the general audience.

3. COMMENT:

> Regarding the scientific content, my view is that the results are interesting and certainly represent a significant progress and should be published in Nature Communications, particularly if the physical interpretation and clarity can be improved. OK, we have learn that there are three main, qualitatively different regimes of intensity fluctuations, below, around and far above the threshold. We also have learn that the observed regimes, in particular the last one, can be explained with statistical distributions that can be interpreted as arising from a hierarchy of multiple time scales. This last bit is very interesting. But, then, the story ends. At least this episode ends. The authors tell us wait for the next episode, where they will attempt a more detailed analysis to be presented in another paper. Thus, in addition to revising the text to improve clarity, the authors are invited to extend on the physical interpretation and implications of the observations being reported here. I do not think the authors need to do additional work, but explain (much) better. I expect that after such a revision, the manuscript should be suitable for publication in Nature Communications.

3. ANSWER: We have deepened the physical interpretation and possible implications of our results within the theoretical framework applied to the experimental data set available at present. For instance, we now elaborate more about the role of the

increasing intensity fluctuations with the excitation power on the mixing ratios of the statistical mixture (see paragraph after Eq. (12) of page 5). In fact, the shift in the weighting parameter in Eq. (11), from $p = 0.94$ near the threshold to $p = 0.30$ at $P/P_{th} = 2.92$, is actually consistent with the increase in the statistical relevance of the component of the mixture that is associated with the largest mean variance, a result that is expected since fluctuations become stronger above the threshold.

A similar discussion on the behavior of the remaining parameters (β , β' , ϵ_{00} , ϵ_{00}') in terms of the statistical properties of the experimental distributions for both $x(t)$ and $\epsilon(t)$ is now provided in the paragraph after Eq. (12) of page 5.

Moreover, we also added a whole new paragraph (penultimate paragraph of page 5 starting with “In this respect...”) that deepens the discussion about the underlying physics of the hierarchy of multiple time scales, particularly on the rising number of relevant scales upon increasing the excitation power. Indeed, near the threshold the emergence of the K-distribution ($N = 1$) may be attributed to the modulation of small-scale fluctuations by more slowly changing large-scale structures. Beyond the threshold, however, stronger nonlinearities may lead to a multiscale dynamics where small-scale intensity fluctuations are modulated by larger scales, and so on up to the largest scale, in a turbulent-cascade-like scenario. The value of N thus represents an estimate of the number of relevant levels in this cascade. In this context, a new reference has been added (ref. [54]) where a similar discussion is provided about the emergence of K-distributions in weak-scattering by continuous media.

We also extended our remark in the concluding paragraph on the possible connection between our findings and the spin-glass phase observed above threshold in the Er-RFL system (see answer in item 9 below).

4. COMMENT:

> Below, I will outline some specific points that relate to the clarity of the message or some minor technical points. They are presented in no particular order:

> - Part of the difficulty arises from the language. Please just simplify it. Most of the readers are not native speakers.

4. ANSWER: We thank Reviewer #2 for this observation. We have rephrased many sentences throughout the revised manuscript in order to make them more concise and understandable to the general audience.

5. COMMENT:

> - The manuscript should be more self-contained. E.g., what is a K-distribution? It is unreasonable to assume every reader knows it. Wiener is better known, but even that still requires some holding of the reader's hand.

5. ANSWER: We have made an effort to identify and explain concepts in the manuscript that are not of regular use by a broad audience. For example, we now

provide some basic remarks on the K-distribution (text after Eq. (1) of page 3) and characterize better the Wiener process (text after Eq. (2) of page 3). Some other changes include the definition of intermittency in the context of turbulence (fourth paragraph of the Results section on page 2) and a discussion on the general meaning of the replica-symmetry-breaking properties of the photonic spin-glass phase observed above the threshold in RL and RFL systems (second paragraph of the Results section). Moreover, a great number of other minor changes were also implemented along the whole manuscript.

6. COMMENT:

> - Some of the terminology is strange for an optics/lasers person: "Intensity spectra" appears without being well defined. If they had said optical spectra, it would have been more understandable. I initially thought that they recorded the intensity over time and then calculated its spectrum (as you would do to analyze, e.g., intensity/amplitude noise). I had to look up the JOSA B paper of the authors to be sure about it. I would simply include a figure showing an example optical spectrum, just like their Figure 2 in the JOSA B paper (reference 35 of the present manuscript).

6. ANSWER: We agree and replaced the term "intensity spectra" either by "optical spectra" or "emission spectra" all over the revised manuscript.

Also, we thank Reviewer #2 for this nice suggestion of including a new figure (Fig. 1 of the revised manuscript), similar to Fig. 2 of the JOSA B paper, showing the optical spectra at the three excitation powers analyzed. We think this new figure will actually help to make our results clearer and more easily understandable to the general audience.

7. COMMENT:

> - The plots need also be more clear. Fig. 2, main figure, vertical axis. It is logarithmic, but there is only one tick mark. There is none other (same problem in Figure 3a). So the scale is unspecified. The same figure's inset: it is the same thing in linear scale, but the data is plotted as bars. This does not match the main figure and confused me before I figured it out from the caption.

7. ANSWER: As mentioned in item 2 above, we proceeded very carefully with a complete revision of all figures. In every case, we checked on the axes, tick marks, symbols, scales, units, labels, captions, etc. The specific points raised above on the previous Figs. 2 and 3 (current Figs. 3 and 4) have been fixed. In particular, Fig. 3 now appears more easily readable after the removal of one of the insets that actually did not add any new information to the reader. Consequently, we thus believe that no place for confusion regarding the figures has been left in the revised manuscript.

8. COMMENT:

> - Same figure, caption: What is an intensity increment? Yes, OK, I understood it

eventually, but please explain things clearly or use more common terminology.

8. ANSWER: The concept of intensity increment is now more clearly explained both in the text (penultimate paragraph of page 2), as well as in the caption of Fig. 3, where its distribution appears plotted for the first time. We also added an explanation in the penultimate paragraph of page 2 on the justification for the use of intensity increments (rather than the maximum intensities themselves) in the analysis of photonic turbulence in Er-RFL.

9. COMMENT:

> - The potentially interesting remark about the spin-glass paper (reference 36) is not clear. I still do not understand it.

9. ANSWER: We tried to make clearer and to extend our remark in the concluding paragraph on the possible connection between our findings and the replica-symmetry-breaking spin-glass phase observed above threshold in the Er-RFL system.

It is actually a striking fact that the emergence of turbulence behavior coincides precisely with the onset of the photonic spin-glass phase at the laser threshold in Er-RFL. In this sense, we first observe that amplification, nonlinearity, and disorder are the essential ingredients to induce both the photonic glassiness and turbulent phenomena in this system. It remains, however, rather elusive whether or not there exists some strict interplay between these properties mediated by the common underlying mechanisms. We cite as example another remarkable finding in similar systems relating features that are also based on the same mechanisms: though photonic glassy properties and Lévy statistics of intensity fluctuations have been also concurrently demonstrated in some RL and RFL systems, it has been recently shown by Tommasi et al. (new ref. [55] in the revised manuscript) that a strict connection between these features is not mandatory, so that there can be circumstances in which, e.g., a glassy phase emerges along with a Gaussian statistics of intensity fluctuations.

Therefore, we thus hope that our work stimulates this unique opportunity to further investigate on these remarkably challenging complex phenomena through controlled photonic experiments in RL and RFL systems.

10. COMMENT:

> - A technical comment: The pump noise is specified to be less than 5% (how much less? we do not know, but presumably not much less than 5% otherwise the authors would have said so) means that at least for 30-40% of the time the laser is above threshold (for the case of $P/P_{th} = 0.99$), since P itself varies by ± 0.05 . So, the laser randomly goes above and then falls below the laser threshold. Surely, this must have consequences ($N=2?$). The authors should remark.

10. ANSWER: As a response to this important technical aspect, we would like to refer again to ref. [35], which, as pointed by Reviewer #2, presents relevant information that

does not need to be repeated here. Indeed, the pump noise is actually around 5% (and not much less than that), and we recall that it is always above threshold. Therefore, this value was constant for all excitation powers analyzed in our work. Referring to Fig. 2(c) in ref. [35], the normalized standard deviation of both the pump laser and Er-RFL was measured, showing clearly that while this quantity remained essentially constant in the pump laser, it varied substantially in the Er-RFL system. Also, this piece of technical information does not have any direct impact on the interpretation of the N values. In this respect, we modified accordingly the corresponding text in the “Intensity measurements” part of the Methods section in the revised manuscript.

11. COMMENT:

> - Can the authors comment on the physical interpretation of the hierarchy number, N ? $N = 2$, perhaps (probably) naively, can be imaged as corresponding to the two time scales associated with the laser being (occasionally) above and then below threshold (related to the previous comment). As P/P_{th} is further increased, N also increases to 6, indicating increasingly complex behavior, developing turbulence. Well, how does N vary as P is increased? Does it increase rapidly to 6 and more or less stay there? What would N be if P/P_{th} was even larger? Does it saturate? Any comments would be appreciated.

11. ANSWER: In our response in item 3 above, we address the question on the underlying physics of the hierarchy of multiple time scales, particularly on the rising number of relevant scales upon increasing the excitation power. As mentioned there, we have added a new paragraph where the physical meaning of the number of scales N is discussed in more detail. We hope that these changes satisfactorily answer the general aspects of the question raised by Reviewer #2. Regarding the more specific points on how rapidly N increases with P , or whether it saturates, we would actually need to investigate on other excitation powers above threshold to comment more properly.

Reviewers' comments:

Reviewer #1 (Remarks to the Author):

Review of the manuscript entitled "Turbulence Hierarchy in a Random Fiber Laser" by González and coworkers: second round.

The authors properly addressed the point I raised in my report by providing a phenomenological explanation of the changes observed in the fitted parameters. I am therefore glad to recommend the publication of this manuscript in Nature Communications.

One small remark only: On page 4, the reference to the panels of Fig. 1 when discussing the time series of maximum intensities appears to be wrong: "shown in Fig. 1(c)-(e)" should be "shown in Fig. 1(d)-(f)".

Reviewer #2 (Remarks to the Author):

The authors have substantially improved the presentation of the manuscript and although I still have several suggestions, I am now supportive of publication:

- The discussion about N is still not satisfactory. I do not find a clear physical explanations on this issue. I understand that the authors are probably unable to give something better at this stage. I urge them to try harder, but if this is not possible, the current level is still sufficient for publication.

- I noticed that physical origin of "nonlinearity" in the laser is not explained at all in the text. Which form of nonlinearity? It is an essential part of the dynamics, so without a doubt deserves a sentence or two. The authors again assume that the readers will have read many of the cited papers, like [42], etc, but this should not be assumed. Most people would normally think of Kerr nonlinearities in a random laser around the threshold of lasing being vanishingly small in magnitude and in effect, so this has to be spelled out.

- I think acronyms should be used sparingly especially since there is no page limit; I would drop RL and RFL. They don't stand in for long expressions, so space economy is not a strong argument and the two of them are similar to each, so this reduces readability of the manuscript, especially if you are not reading it line by line from beginning to end. This is just a suggestion.

RESPONSE TO THE REVIEWER #2 – SECOND ROUND

Re: Manuscript “Turbulence hierarchy in a random fiber laser”, by I. R. R. González et al. Code NCOMMS-17-02149A.

We thank once again Reviewer #2 for the careful and comprehensive reading of our manuscript in this second round. His/Her comments and suggestions evidence a genuine concern in making the manuscript more accessible to a broad readership, while turning, at the same time, the physical discussion on the results as thorough as possible.

We are glad that the changes made in the first round “have substantially improved the presentation of the manuscript”, according to Reviewer #2. Nevertheless, we still have made an effort to properly address the new suggestions, as detailed below.

1. COMMENT:

> - The discussion about N is still not satisfactory. I do not find a clear physical explanations on this issue. I understand that the authors are probably unable to give something better at this stage. I urge them to try harder, but if this is not possible, the current level is still sufficient for publication.

1. ANSWER: As Reviewer #2 anticipated, it has been extremely difficult to elaborate more deeply on this rather subtle issue with the experimental data set available at present. Perhaps a detailed investigation on the behavior of N with the increasing excitation power will allow us to provide more precise statements on the hierarchy of multiple time scales, without risking being too much speculative. Therefore, in this respect we have just opted to add a sentence in the first paragraph of page 6 claiming that the existence of a hierarchy of time scales has been clearly identified with basis on the statistical analysis of a large data set of intensity increments, though more studies are still necessary to understand more comprehensively the origin of the underlying physical mechanisms.

2. COMMENT:

> - I noticed that physical origin of "nonlinearity" in the laser is not explained at all in the text. Which form of nonlinearity? It is an essential part of the dynamics, so without a doubt deserves a sentence or two. The authors again assume that the readers will have read many of the cited papers, like [42], etc, but this should not be assumed. Most people would normally think of Kerr nonlinearities in a random laser around the threshold of lasing being vanishingly small in magnitude and in effect, so this has to be spelled out.

2. ANSWER: Reviewer #2 is right. Although in the first round we have been concerned in explaining to the wide audience the meaning of relevant underlying ingredients (e.g., intermittency), we acknowledge that understanding the origin of nonlinearity in the Er-RFL system is also essential. We thus added an explanation in the last paragraph of page 1 of the revised manuscript, right after the sentence “We show that the interplay of

nonlinearity and disorder is essential to induce photonic turbulence in the distribution of intensity fluctuations in the RL phase of Er-RFL.” We now state that the microscopic origin of the nonlinearity in Er-RFL actually lies in the electronic levels of the erbium ions, being evidenced through the process of single photon induced nonlinear absorption. Moreover, at this point we also added for completeness that the origin of disorder in this system is due to the random fiber gratings inscribed in the doped fiber.

3. COMMENT:

> - I think acronyms should be used sparingly especially since there is no page limit; I would drop RL and RFL. They don't stand in for long expressions, so space economy is not a strong argument and the two of them are similar to each, so this reduces readability of the manuscript, especially if you are not reading it line by line from beginning to end. This is just a suggestion.

3. ANSWER: We thank Reviewer #2 for this suggestion. We have now dropped the acronyms RL and RFL all over the manuscript, and we feel that this change was indeed positive.

The only acronym that we still keep is Er-RFL, since we think that the many repetitions of the lengthy expression “erbium-based random fiber laser” would tire the reader unnecessarily.